# TOKEN-BY-TOKEN ELECTION: IMPROVING LANGUAGE MODEL REASONING THROUGH TOKEN-LEVEL MULTI-MODEL COLLABORATION

## ABSTRACT

With the continuous development of large language models (LLMs), they have demonstrated amazing capabilities in many areas of natural language processing (NLP). However, due to their inherent limitations, the performance of a single model on many complex reasoning tasks has reached a bottleneck. A feasible solution is to introduce external feedback to further improve model performance, among which multi-model collaboration is a particularly promising approach. In this paper, we propose token-by-token election (TTE), a novel token-level multi-model collaboration strategy. Different from the common multi-model collaboration methods that operates at the overall answer level, TTE performs multi-model elections at the lowest token level. It selects the optimal token from the next token distributions given by multiple LLMs and then generates the answer autoregressively, allowing multiple LLMs to reach a consensus on each token. Inspired by human behavior, TTE consists of three election modes, including Cooperation, Competition, and Counting, all of which aim to sample the optimal token from multiple distributions. By strictly controlling the generation quality of each token, TTE can improve the quality of the overall answer and break through the performance bottleneck of a single LLM. Through extensive experiments on a variety of different types of reasoning benchmarks, we demonstrate the powerful performance of TTE, which further improves the performance compared to the current state-of-the-art single LLM and other multi-model collaborative methods. The code will be released on GitHub.

## 1 INTRODUCTION

With the rapid development of large language models (LLMs), numerous impressive works such as GPT4 (Achiam et al., 2023), Llama3 (Dubey et al., 2024), and Qwen2 (Yang et al., 2024) have emerged. People are increasingly accustomed to seeking answers from LLMs when encountering problems, and even researchers consult LLMs during their scientific work. Although LLMs have demonstrated remarkable capabilities in many areas of natural language processing (NLP), they often show their inability to perform complex reasoning tasks (Fu et al., 2022). Therefore, how to further improve the performance of LLMs in complex reasoning tasks has become a hot topic (Kojima et al., 2022; Liang et al., 2023). Enhancing model performance from training side is very costly, as training a language model requires significant resources. Furthermore, performance improvements have begun to plateau due to the slowing impact of scaling laws (Kaplan et al., 2020; Touvron et al., 2023). Therefore, more and more research (Wei et al., 2022; Wang et al., 2022; Kojima et al., 2022; Madaan et al., 2024; Shinn et al., 2024) has begun to focus on improving model performance from the inference side.

Improving the performance of LLMs from the inference side differs from training-based methods which optimize model parameters (Ouyang et al., 2022; Rafailov et al., 2024). It is typically achieved by optimizing the model input (Zhang et al., 2022; Kojima et al., 2022) or refining the model output (Madaan et al., 2024; Kim et al., 2024), so it generally does not require additional training data and computing resources. Two typical methods are the chain of thought (CoT) approaches (Wei et al., 2022; Fu et al., 2022) and the self-correction approaches (Madaan et al., 2024; Shinn et al., 2024). The former improves response quality by providing several examples to guide LLMs in answering

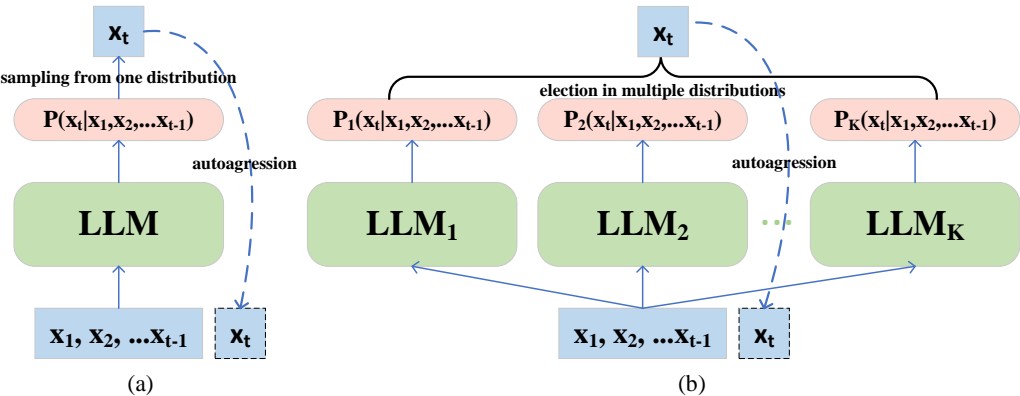

Figure 1: From the implementation point of view, the proposed TTE is an adjustment based on single model autoregression. (a) A single LLM samples the next token from its output next token distribution and generates the response autoregressively. (b) The proposed TTE selects the optimal next token from multiple next token distributions given by multiple LLMs and generates the response autoregressively, realizing multi-model collaboration at the lowest token level.

questions step-by-step, while the latter enhances response quality through iterative refinement of the answers. However, as the performance of individual LLMs continues to improve (Dubey et al., 2024; GLM et al., 2024), recent studies (Huang et al., 2023; Kojima et al., 2022; Yin et al., 2023) have pointed out that the performance gains of these two methods on LLMs are diminishing, particularly for smaller models. In addition, these two methods that rely on the model's own capabilities to improve the quality of answers are difficult to break through the inherent limitations of the model without external feedback. An effective solution is to introduce external feedback to further improve model performance, among which multi-model collaboration is a very promising approach.

Different LLMs usually have different knowledge boundaries and their own strengths (Wan et al., 2024; Jiang et al., 2023). If their capabilities can be well synergized, it will certainly enhance the reasoning ability of the language model and break through the performance bottleneck of a single LLM (Khan et al., 2024; Du et al., 2023), which is also the expected goal of multi-model collaboration. Previous multi-model collaboration methods (Khan et al., 2024; Liang et al., 2023; Du et al., 2023) mostly conduct majority voting or discuss at the level of the overall answer. For the same question, each LLM puts forward its own point of view and tries to convince other models, so as to finally reach a consensus among multiple models. The debate process usually involves selecting the final answer based on certain rules or introducing a new referee model. This may require many rounds of debate and relies heavily on one of the models to give a strong correct answer and convince the other debaters to get the final answer.

It is generally believed that the knowledge of LLM is stored in its massive parameters, but the output next token distribution is the specific external manifestation of its knowledge [1] (Hinton, 2015; Wan et al., 2024; Radford et al., 2019). **Therefore, a straightforward idea is to combine the knowledge of multiple LLMs by combining the next token distributions given by these models**. In this paper, we propose token-by-token election (TTE), a novel token-level multi-model collaboration strategy. Specifically, TTE generates answers based on autoregression (Vaswani, 2017; Radford, 2018), but unlike the autoregression of a single LLM, the token generated by TTE at each step is sampled from multiple next token distributions generated by different LLMs. Since each token is generated through the collaboration of multiple models, the final answer is also produced through their joint decision. Huang et al. (2024) have pointed out that some low-quality answers may start to deteriorate from a certain token. Our starting point is to improve the quality of the overall answer by improving the quality of each token. We believe that the tokens selected by the joint decision of multiple models are better. This behavior is similar to a review team checking the document word by word to avoid errors and discussing uncertain parts to determine the final wording thereby improving the quality of document. Moreover, we are surprised to find that this token-level collaboration strategy even exhibits some emergent capabilities. In some cases, TTE can give the

---

[1]LLM's output is obtained by autoregressive sampling from this distribution, so LLM's knowledge determines the quality of its output.

correct answers even when multiple LLMs individually give incorrect answers, which reflects the saying "two heads are better than one".

In order to sample the optimal token from the next token distributions given by multiple LLMs, inspired by human behavior, we design three sampling modes: (1) Cooperation, comprehensively considering the next token distributions given by multiple LLMs to make a selection; (2) Competition, competing among the candidate tokens given by multiple LLMs to make a selection; (3) Counting, counting votes for different tokens from multiple LLMs to make a selection. These three election modes are all dedicated to selecting the optimal token from the next token distributions provided by multiple LLMs, strictly controlling the quality of each step of autoregression to improve the overall answer quality, thereby enhancing the language model reasoning.

We conduct extensive experiments on three categories of tasks: mathematical reasoning, symbolic reasoning, and commonsense reasoning. The results show that our approach achieves significant improvements over previous strong baselines. In addition, each mode performs well in different categories of tasks, demonstrating its specific strengths. Further exploration reveals the advantages of TTE over existing multi-model methods, achieving our goal of combining the capabilities of different models, and even demonstrating certain emergent capabilities (Lu et al., 2023). We also conduct an evaluation on a reliability evaluation benchmark, and the results show that TTE can improve the authenticity of the answer to a certain extent, thereby enhancing the credibility of decisions.

In summary, our main contributions include: (1) We propose token-by-token election, a novel token-level multi-model collaboration strategy. As far as we know, we are the first to attempt multi-model collaboration at the token level, offering a new perspective in this field. (2) We introduce three election modes, including cooperation, competition and counting, all of which are committed to improving the quality of the selected token, thus breaking through the performance bottleneck of a single LLM. (3) Evaluations on multiple reasoning benchmarks demonstrate the superior performance of our proposed method.

## 2 TOKEN-BY-TOKEN ELECTION

Given the same question, different LLMs usually give different answers. This is because they have learned different knowledge due to the difference in network architecture, training data and training process (Raiaan et al., 2024). As mentioned earlier, it is generally believed that the knowledge is stored in the huge parameters of LLMs (Radford et al., 2019), and the output next token probability distribution is the specific external manifestation of their knowledge (Hinton, 2015; Wan et al., 2024). Therefore, our starting point is to combine the knowledge of different LLMs by combining these distributions:

$$
\begin{aligned}
P_t &= Combine(P_t^1, P_t^2, ..., P_t^K), \\
KN_t &= Combine(KN_t^1, KN_t^2, ..., KN_t^K),
\end{aligned}
\tag{1}
$$

where $P_t$ represents distributions, $KN$ represents knowledge and $Combine$ represents a certain combination.

The primary objective of the proposed TTE is to select the optimal token at each step of autoregression through multi-model collaboration, thereby improving the quality of the overall output by enhancing the quality of individual tokens, since the output of the language model is composed of these individual tokens. The overall architecture of TTE is shown in the Figure 1. Specifically, we aim to combine the different next token probability distributions generated by multiple LLMs to select the current optimal token. As these different distributions encapsulate distinct knowledge from the respective LLMs, our aim is to achieve the optimal integration of diverse LLM capabilities in this manner. By incorporating external feedback and knowledge from various LLMs (Yin et al., 2023), our multi-model collaboration method TTE can overcome the performance limitations of a single model.

In the following sections, we first introduce some basic content of LLM autoregression, then introduce the three election models we proposed, and finally briefly summarize the differences between our method and previous methods.

## 2.1 PRELIMINARIES

From the implementation point of view, the proposed TTE is an adjustment based on single model autoregression, as shown in Figure 1. For single LLM inference, given an input text sequence $Text$, it will first be tokenized to obtain a token sequence $T$:

$$T = tokenize(text), \tag{2}$$

where $T = \{t_i\}_{i=1}^N$, $t_i$ represents an independent token in $T$, and $N$ is the length of $T$. The tokenize operation is a necessary step for LLM inference. It converts the text into a sequence of tokens that the LLM can process. This is usually done by a tokenizer equipped with the LLM. Then we feed $T$ into an LLM $\mathcal{M}$, which will calculate the probability distribution $P$ of the next token, $P$ is essentially a conditional probability distribution:

$$P(t_{N+1}|\{t_i\}_{i=1}^N) = \mathcal{M}(\{t_i\}_{i=1}^N), \tag{3}$$

$P$ contains the probability values for each token in the LLM tokenizer's vocabulary being the next token. The next token will be sampled based on $P$ and will be used as additional input to the original sequence to continue generating subsequent tokens until the end token $< eos >$ (end of sentence) is output. At each step the model is auto-regressive (Graves, 2013), consuming the previously generated symbols as additional input when generating the next (Vaswani, 2017). Finally, the tokenizer will decode all generated tokens back into text to get the final output result of LLM.

## 2.2 ELECTION MODES

Given the same input, $K$ LLMs $\{\mathcal{M}_j\}_{j=1}^K$ will output $K$ next token distributions $\{P_j\}_{j=1}^K$. Since the tokenizers of different LLMs are different, we need to decode them into the probability distribution of the next word so that the $K$ LLMs can be aligned in the same semantic space. So we can get $K$ next word distributions $\{P_j^w\}_{j=1}^K$. What we need to do is to sample the optimal next word $next\_word$ from $\{P_j^w\}_{j=1}^K$:

$$next\_word = Sample(\{P_j^w\}_{j=1}^K). \tag{4}$$

For sampling from a single distribution, strategies such as Top-k (Fan et al., 2018) and Top-p (Holtzman et al., 2019) are commonly used. For sampling from multiple distributions, inspired by human behaviors, we have designed the following three sampling modes, including Cooperation, Competition and Counting:

**Cooperation.** For each next word probability distribution $P^w$, we use the Top-k (Fan et al., 2018) strategy to sample the top $H$ words with the highest probability (the probability of the other words is very small so there is no need to consider them, which can also reduce complexity). In this way, from the $K$ distributions given by $K$ LLMs, we can get $K$ dictionaries, where each key represents a word, and each value represents its probability. Cooperation is to add the values of these $K$ dictionaries according to the same key, and then select the one with the highest probability. In this way, the optimal next word is elected through multi-model cooperation.

**Competition.** For each next word probability distribution $P^w$, we select the word with the highest probability as the candidate, so we get $K$ candidates from $K$ LLMs. It is generally believed that the higher the probability of the LLM output, the greater its confidence in that output (Radford et al., 2019; Liang et al., 2022; Kuhn et al., 2023; Shih et al., 2023). We select the candidate with the highest probability as the optimal next word, which is equivalent to selecting the LLM with the highest confidence to answer at each step. In this way, the optimal next word is elected through multi-model competition.

**Counting.** For each next word probability distribution $P^w$, similar to the cooperation mode, we only keep the top $H$ words, which all have the probability of being the next word Fan et al. (2018); Jiang et al. (2021). We count the number of times each word appears in the $K$ dictionaries. The higher the count, the more LLMs we believe recognize it. We select the word with the most appearances as the optimal next word. If two words have the same number of votes, we select the one with a higher total probability. In this way, the optimal next word is elected through counting the votes given by multiple LLMs.

The pseudo code of Algorithm 1 shows the complete process of TTE generating an answer based on a question, note that we have only described the general core process and many details have been omitted. We also give a theoretical analysis of TTE in Appendix A.

---

**Algorithm 1** Token-by-Token Election

---

**Require:** Multiple LLMs $\{\mathcal{M}_j\}_{j=1}^K$, query $q$

0 : $response = str()$, $length = 0$, storing answers and indicating answer length

1 : **while** $length <= max\_length$:

2 :      for $j$ in range($K$):

3 :          Input query $q$ into LLM and get the next token distribution: $P_j = \mathcal{M}_j(q)$.

4 :          Convert the next token distribution $P_j$ to next word distribution $P_j^w$.

5 :      Use the election modes to sample the next word: $next\_word = Sample(\{P_j^w\}_{j=1}^K)$.

6 :      Splice the $next\_word$ as additional input to the original input: $q = q.join(next\_word)$.

7 :      Sequence length increases: $length = length + 1$.

8 :      Generate answers autoregressively: $response = response.join(next\_word)$.

9 :      Determine whether the end token $< eos >$ is generated, if so, stop the loop.

10 : **end**

11 : **return** $response$

---

## 2.3 RELATION TO PRIOR WORK

FuseLLM (Wan et al., 2024) also leverages the distributions provided by multiple LLMs, but it is a training-based method aimed at knowledge distillation. It combines the next token probability matrices given by multiple LLMs to obtain soft labels, which are used to supervise the training process of the student model, thereby distilling the knowledge from multiple teacher models into the student model. In contrast, our TTE directly samples from multiple distributions and generates answers autoregressively, making it a method that does not require training. The two approaches have fundamental differences.

## 3 EXPERIMENTS

### 3.1 EXPERIMENT SETUP

**Tasks ans datasets**. We evaluate the performance of our proposed TTE method on the following benchmarks:

- **Arithmetic Reasoning**. For this task, we selecte four datasets of different difficulty, including SVAMP (Patel et al., 2021), GSM8K (Cobbe et al., 2021), AddSub (Hosseini et al., 2014) and AQuA (Ling et al., 2017).

- **Commonsense Reasoning**. We select four datasets to evaluate the TTE's performance, including CommonseQA (Talmor et al., 2018), StrategyQA (Geva et al., 2021), OpenBookQA (Mihaylov et al., 2018) and ARC-c (Clark et al., 2018).

- **Symbolic Reasoning**. We select four datasets from BigBench (Srivastava et al., 2022) for testing, including Date Understanding, Penguin, Colored Objects and Logical Deduction.

- **Reliability Evaluation**. We select TruthfulQA (Lin et al., 2021) to evaluate the reliability and truthfulness of the answers generated by language models.

**Baselines**. We compare the proposed TTE with three sets of widely used baselines: (1) source LLMs, including Qwen-2 7B (Yang et al., 2024), Llama-3 8B (Dubey et al., 2024) and GLM-4 9B (GLM et al., 2024); (2) CoT (Wei et al., 2022) methods: the performance of the above three LLMs after using CoT prompting; (3) Majority Voting, a widely used collaboration method, the answers of the three models are subjected to majority voting to select the most consistent answer. We also compare with two multi-model collaboration methods EoT (Yin et al., 2023) and MAD (Du et al., 2023) on partial datasets.

**Implementation details**. In the main experiment, we use three SOTA open source LLMs: Qwen-2 7B, Llama-3 8B and GLM-4 9B. The top 5 words are sampled in the Top-k sampling algorithm. We use regular expressions to extract the answers from LLM's answers to calculate the accuracy. The whole method is training-free, and most experiments are completed on one Nvidia H800 GPU.

Table 1: Comparison of accuracy on four mathematical reasoning datasets using various election modes of TTE and strong baselines. The best results are highlighted in **bold**. All results are expressed as a percentage of accuracy, with the % symbol omitted. We also compute the arithmetic average of the four results for quick comparison, which is shown in the Avg. column.

| Methods / Datasets | | SVAMP | GSM8K | AddSub | AQuA | Avg. |
|---|---|---|---|---|---|---|
| Single LLM | Qwen-2-7B | 88.8 | 81.5 | 90.3 | 61.4 | 80.5 |
| | Llama-3-8B | 80.6 | 77.5 | 84.3 | 48.4 | 72.7 |
| | GLM-4-9B | 86.2 | 77.7 | 87.6 | 55.9 | 76.8 |
| CoT | Qwen-2-7B | 90.0 | 82.3 | 90.8 | **65.5** | 82.1 |
| | Llama-3-8B | 85.2 | 79.6 | 86.5 | 54.2 | 76.3 |
| | GLM-4-9B | 88.6 | 79.6 | 87.5 | 58.1 | 78.4 |
| Multi-Model | Majority Voting | 90.8 | 79.0 | 91.3 | 60.5 | 80.4 |
| TTE (Ours) | Cooperation | 91.2 | 83.3 | **91.5** | **65.5** | 82.8 |
| | Competition | **91.5** | **84.9** | 91.3 | 64.1 | **83.0** |
| | Counting | 90.3 | 84.1 | 90.3 | 60.3 | 81.2 |

Table 2: Comparison of accuracy on four commonsense reasoning datasets using various election modes of TTE and strong baselines.

| Methods / Datasets | | CSQA | StrategyQA | OpenBookQA | ARC-c | Avg. |
|---|---|---|---|---|---|---|
| Single LLM | Qwen-2-7B | 68.3 | 69.6 | 78.0 | 79.7 | 73.9 |
| | Llama-3-8B | 65.4 | 68.8 | 71.4 | 74.2 | 70.0 |
| | GLM-4-9B | 65.2 | 71.3 | 79.6 | 80.5 | 74.4 |
| CoT | Qwen-2-7B | 71.9 | 73.2 | 81.0 | 81.0 | 76.7 |
| | Llama-3-8B | 67.9 | 70.2 | 75.3 | 76.3 | 72.4 |
| | GLM-4-9B | 67.3 | 71.8 | 79.5 | 79.4 | 74.5 |
| Multi-Model | Majority Voting | 72.2 | 71.8 | 80.6 | 83.2 | 77.0 |
| TTE (Ours) | Cooperation | **76.9** | 75.2 | 82.2 | 83.1 | 79.3 |
| | Competition | 75.7 | 74.8 | 80.2 | 82.1 | 78.2 |
| | Counting | 76.5 | **75.6** | **83.4** | **83.8** | **79.8** |

## 3.2 MAIN RESULTS

We select Qwen-2 7B, Llama-3 8B, and GLM-4 9B as the baseline single LLMs, which are among the most advanced open-source LLMs. Each of these models has its own advantages and disadvantages in different datasets and generally represents the best performance of models with similar parameter scales on these datasets.

**Mathematical Reasoning.** The quantitative results on four datasets are shown in Table 1. It can be seen that the proposed TTE achieves significant performance improvements in most cases. Comparing the three single models, Qwen performs the best in mathematics, while Llama performs the worst. It can also be observed that the performance improvement brought by CoT to Llama is significantly greater than that of the other two models, but CoT does not bring much improvement to the other two models with better performance. Furthermore, naive majority voting does not guarantee performance improvement over the best-performing single model. Finally, all three collaborative modes are improved compared to the single model, and the Competition mode performs the best in terms of average scores.

**Commomsense Reasoning.** The quantitative results on four datasets are shown in Table 2. It can be clearly seen that the proposed TTE achieves significant performance improvements. Comparing the three single models, GLM performs the best in this task, while Llama still performs the worst. It is observed that that the performance improvement brought by CoT is still not significant enough, especially for the GLM, there is almost no improvement. In addition, majority voting brings certain performance improvements, which are generally better than the best performance of a single model. Finally, all three collaborative modes are improved compared to the single model, and the Counting mode performs the best in terms of average scores.

**Symbolic Reasoning.** The quantitative results on four datasets are shown in Table 3. The proposed TTE still achieves the best performance in most cases, with the Cooperation mode performing best and the Competition mode performing worse in terms of average scores, but all of them are improved compared to the best performance of a single LLM. Comparing the three single models, GLM

Table 3: Comparison of accuracy on four symbolic reasoning datasets from BigBench using various election modes of TTE and strong baselines.

| Methods / Datasets | | Date | Penguin | Colored Objects | Logical Deduction | Avg. |
|---|---|---|---|---|---|---|
| Single LLM | Qwen-2-7B | 57.2 | 77.4 | 74.0 | 74.8 | 70.9 |
| | Llama-3-8B | 65.6 | 75.3 | 64.8 | 63.6 | 67.3 |
| | GLM-4-9B | 54.0 | 81.5 | 76.4 | 81.4 | 73.3 |
| CoT | Qwen-2-7B | 66.5 | 79.2 | 73.2 | 77.3 | 74.1 |
| | Llama-3-8B | **69.6** | 76.1 | 66.4 | 70.1 | 70.6 |
| | GLM-4-9B | 58.2 | 81.3 | 76.9 | 82.0 | 74.6 |
| Multi-Model | Majority Voting | 65.0 | 82.3 | 79.5 | 81.2 | 77 |
| TTE (Ours) | Cooperation | 68.9 | **84.9** | 78.8 | **83.2** | **79.0** |
| | Competition | 68.5 | 82.8 | 76.2 | 78.4 | 76.5 |
| | Counting | 67.3 | 81.5 | **81.2** | 82.4 | 78.1 |

performs the best in this task, while Llama still performs the worst. Moreover, it can be seen that the performance improvement brought by CoT is still not significant enough and majority voting brings certain performance improvements, which are generally better than the best performance of a single model.

**Analysis**: From the above experiments, it can be seen that compared with the performance of single models and other baseline methods, the three modes of TTE perform relatively well and have improved. In general, the three modes have their own excellent performances. The Cooperation mode performs best in symbolic reasoning tasks, the Competition mode performs best in mathematical reasoning tasks, and the Counting mode performs best in commonsense reasoning tasks.

In addition, it can be clearly seen that CoT does not improve the performance of a single model enough in most cases and sometimes even causes performance regression. A possible explanation is that more chain thinking data has been introduced in the post-training (Rafailov et al., 2024; Ouyang et al., 2022) process of these latest LLMs (Dubey et al., 2024; Yang et al., 2024; GLM et al., 2024) so they have learned to think step by step even without obvious prompts. Besides, some previous studies (Wei et al., 2022; Kojima et al., 2022) have also shown that the CoT method is more effective for LLMs with larger parameters and has less impact on LLMs with fewer parameters.

Finally, we can see that although majority voting is a very simple way of collaboration, it can still bring certain performance improvements compared to single models most of the time, which is also consistent with the experience of human collective wisdom.

## 3.3 RELIABILITY EVALUATION

The validity of LLM outputs is crucial, and their reliability is equally indispensable. The main experiments in Section 3.2 have already demonstrated that our proposed TTE can significantly enhance the reasoning performance of language models. Therefore, we are also interested in exploring whether TTE can improve the reliability of language models. For this purpose, we select TruthfulQA (Lin et al., 2021), a classic and widely used benchmark for evaluating the truthfulness of language model outputs, which can reflect the reliability of LLM outputs to some extent.

The results are shown in Table 4. The three metrics in the table indicate higher truthfulness of the model outputs when they are larger. It can be clearly seen that the metrics for the three single models are relatively close, while our proposed TTE in its three modes achieves significantly higher metrics, indicating that TTE enhances the reliability of language model outputs to some extent. This evaluation of truthfulness differs from the evaluation of reasoning performance, it requires the model to have encountered the relevant content in the training data to answer factual questions, rather than being able to generalize like reasoning tasks. For example, knowing that the previous U.S. president was Trump does not necessarily mean the model can infer that the current president is Biden. The model can only make the correct response to factual questions if it has been trained on relevant data. The knowledge boundary of LLMs largely determines the truthfulness of their responses. The performance of TTE also meets our expectation of effectively combining the knowledge of multiple models to some extent.

Table 4: Performance of TTE in reliability evaluation. We report three metrics on the TruthfulQA benchmark, where larger values indicate more truthful results.

| Model/Metric | | ROUGE | BLEU | BLEURT |
|---|---|---|---|---|
| Single LLM | Qwen-2 7B | 0.548 | 0.532 | 0.659 |
| | Llama-3 8B | 0.555 | 0.517 | 0.634 |
| | GLM-4 9B | 0.587 | 0.536 | 0.660 |
| TTE (Ours) | Cooperation | 0.598 | 0.551 | 0.661 |
| | Competition | 0.593 | 0.546 | 0.657 |
| | Counting | 0.595 | 0.554 | 0.658 |

Table 5: Ablation on the number of used LLMs. The Cooperation mode is used here for multi-model collaboration.

| Model | SVAMP | CSQA | Penguin |
|---|---|---|---|
| Qwen | 88.8 | 68.3 | 77.4 |
| Llama | 80.6 | 65.4 | 75.3 |
| GLM | 86.2 | 65.2 | 81.5 |
| Qwen+Llama | 90.2 | 73.7 | 85.1 |
| Qwen+GLM | 91.0 | 74.1 | 85.3 |
| Llama+GLM | 87.3 | 71.7 | 84.2 |
| Qwen+Llama+GLM | 91.2 | 76.9 | 84.9 |

Table 6: Comparison with other collaboration methods.

| Method | SVAMP | CSQA | Penguin |
|---|---|---|---|
| EoT | 88.9 | 74.3 | 82.1 |
| MAD | 90.1 | 73.9 | 80.3 |
| TTE (Cooperation) | 91.2 | 76.9 | 84.9 |
| TTE (Competition) | 91.5 | 75.7 | 82.8 |
| TTE (Counting) | 90.3 | 76.5 | 81.5 |

## 3.4 ABLATION STUDY

**Ablation on the number of used LLMs.** We show the evaluation results of using different numbers of LLMs in Table 5. It can be seen that as the number of LLMs increases from one to three, the overall performance gradually improves. The magnitude of this change is different in different tasks. We also notice that in some cases, the effect of using two models is better than that of using three. We think this is reasonable. When human groups collaborate, it is not always the case that more participants lead to better outcomes, the ability of the participants is a key factor. But in general, using more models will expand the knowledge boundary of TTE and thus improve performance.

**Ablation of the number of sampled words $H$.** We show the evaluation results of using different numbers of sampled words in Figure 2. The performance of the Competition mode is independent of the number of sampled words. In the Cooperation mode, it can be clearly seen that the performance hardly changes after the number of sampled words reaches 5. The possible reason is that the probability of the top 5 words is large, and the probability of the following words is too small, so their addition will not have much impact on the performance. In Counting mode, as the number of sampled words increases, the performance drops significantly. This is because in counting mode, we prioritize counting the number of times a word appears in multiple distributions when counting votes rather than considering their probability values. When there are too many sampled words, the probability of the following words is actually very small. Introducing too many words with very small probabilities will have a significant negative impact on the results.

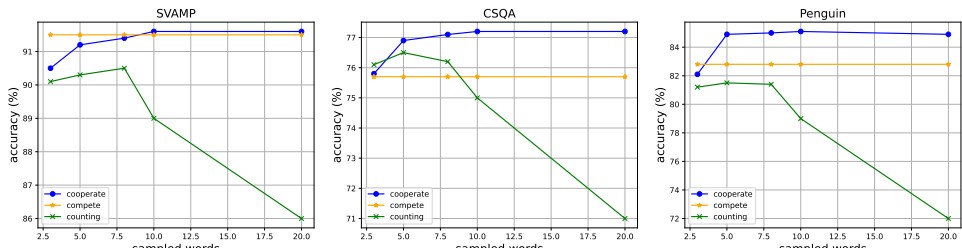

Figure 2: The effect of the number of sampled words on the three modes of TTE.

## 3.5 COMPARISON WITH OTHER COLLABORATION METHODS

As shown in the previous experimental results, our TTE breaks through the performance bottleneck of a single model and achieves the goal of multi-model collaboration. In addition, we also want to explore its competitiveness in multi-model collaboration methods. We select two represen-

Table 7: Examples where TTE is able to answer correctly even though all three models answer incorrectly. Incorrect answers are shown in red and correct answers are shown in blue. Since some answers are too long, we have made some adjustments to facilitate presentation, but the core content remains unchanged. Cooperation mode is used here.

| | GSM8K | CSQA |
|---|---|---|
| **Question** | *The book is printed on paper that, when stacked, is 100 pages to the inch. Each paper is printed on both sides, with one page of the book printed on each side. How many pages are in the book, if it is 1.5 inches thick?* | *Kramer wrote a self-referential book. What might that book be about? (A) counter (B) coffee table (C) school room (D) backpack (E) bedside table* |
| **Qwen** | To find out how many pages are in the book, we need to understand that each inch of the book's thickness represents 100 pages (since the paper is 100 pages per inch). Given that the book is 1.5 inches thick, we can calculate the total number of pages as follows: Total pages = Thickness of the book in inches × Pages per inch Total pages = 1.5 × 100 = 150 Therefore, the book has **150 pages**. | Analysis: A self-referential book is one that refers to itself, often discussing its own structure, content, or writing process. Given this context, the book would likely delve into themes related to literature, writing, or the act of creating a book. Answer: (E) bedside table |
| **Llama** | The book is 1.5 inches thick. Each inch has 100 pages. So 1.5 inches will have 1.5 x 100 = 150 pages. The answer is 150. | Analysis: The question is asking what Kramer might write a self-referential book about. A self-referential book is a book that refers to itself, often in a humorous or ironic way. Kramer is a character from the TV show "Seinfeld", known for his eccentricities and witty remarks. Correct answer: (A) The Kramer |
| **GLM** | If the paper is 100 pages to the inch and each paper is printed on both sides, then each inch of thickness corresponds to 50 pages (since one side of the paper is one page). If the book is 1.5 inches thick, then the number of pages in the book would be: 1.5 inches * 50 pages per inch = 75 pages. So, the book has 75 pages. | Among the options, "bedside table" (E) is the closest to a book, as it is a piece of furniture that could be associated with a book (books are often placed on bedside tables). Given the limitations of the options, the most appropriate answer would be: (E) bedside table |
| **TTE** | If the book is 1.5 inches thick and each paper is 1/100 of an inch thick, then the number of papers in the book is 1.5 inches * 100 papers/inch = 150 papers. Since each paper has two sides, and each side is a page, the total number of pages in the book is 150 papers * 2 sides/paper = 300 pages. | Coffee table could be a possibility, as a coffee table is a common piece of furniture that Kramer might have written about in a self-referential manner. Answer: (B) coffee table |

tative approaches: (1) Multi-Agent Debate (Du et al., 2023), where multiple models express their arguments, and a judge manages the debate process to determine the final solution. (2) Exchange of Thought (Yin et al., 2023), a recently released method that builds cross-model communication strategies based on network topologies. As shown in Table 6, TTE steadily outperforms these two innovative baselines across different modes.

## 3.6 EXAMPLES OF EMERGENT CAPABILITIES

The goal of TTE is to combine the capabilities of multiple models to break through the performance bottlenecks of a single LLM. Traditional multi-model collaboration methods rely on most models providing the same correct answer for majority voting, or rely on one model providing a highly confident correct answer to convince the others to reach a consensus. However, we are surprised to find that our TTE can provide the correct answer even when each individual model answers incorrectly, demonstrating a certain level of emergent ability, akin to the saying "two heads are better than one".

Typical examples are shown in Table 7. It can be clearly seen that for the same math problem, the three LLMs give wrong answers of 150, 150 and 75 respectively, ignoring the fact that a piece of paper has two sides or confusing the logical relationship, while TTE gives the correct answer of 300; for another question, the three single LLMs fail to figure out the meaning of "a self-refine book" and thus can not give a correct answer, but TTE understands and gives the correct answer. We speculate that the token-level collision of ideas in TTE has brought some different inspirations, and may sometimes achieve the effect of $1 + 1$ being greater than 2. More examples and discussion can be seen in Appendix B.

## 4 RELATED WORK

In this section, we will introduce some mainstream methods to improve LLM performance from the inference side. These methods are all training-free and do not require a lot of computing resources, including chain of thought prompting, self-correction, and multi-model collaboration methods.

## 4.1 CHAIN OF THOUGHT PROMPTING

Wei et al. (2022) first proposed the method to improve the reasoning performance of LLM by constructing chain of thought prompts, mainly by constructing contextual examples to teach LLM to analyze step by step and then output the answer. Since then, many works (Kojima et al., 2022; Wang et al., 2022; Fu et al., 2022; Chia et al., 2023; Zhang et al., 2022) have been further improved on this basis. In addition to the correct examples, Chia et al. (2023) also added the wrong thinking process to let the model further learn to avoid mistakes. Kojima et al. (2022) proposed a zero-shot CoT method, which allows the model to learn step-by-step analysis by adding the universal instruction "Let's think step by step", thus avoiding the need to construct examples. Furthermore, self-consistency method (Wang et al., 2022) was proposed to replace the greedy decoding strategy in CoT, by sampling the outputs of multiple paths and using majority voting to select the final answer, the reasoning performance of the single model was further improved. Subsequently, (Fu et al., 2022) pointed out that more complex thought chain prompts have better effects, and the more complex answers output are more credible, and further improved the self-consistency method based on complexity.

## 4.2 SELF-CORRECTION

A method that utilizes the model's own capabilities for self-correction (Madaan et al., 2024; Shinn et al., 2024; Sean et al., 2022; Kim et al., 2024; Bai et al., 2022) has been proposed to further improve the quality of answers output by the model. Madaan et al. (2024) used the model's own feedback to find errors and modify their own output, improving the quality of the output through continuous iteration and modification until the stopping condition is met. Kim et al. (2024) introduced some simple prompts to allow LLM to recursively criticize and improve its output. Shinn et al. (2024) reflected the task feedback signal through verbal feedback and then maintained the LLM's own reflected text in the episodic memory buffer to induce better decisions in subsequent trials. However, Huang et al. (2023) pointed out that these self-correction methods have great limitations and do not significantly improve LLM in a fair setting.

## 4.3 MULTI-MODEL COLLABORATION

Using multiple LLMs to solve problems is still in its early stages (Yin et al., 2023). It is usually done by imitating some group collaboration behaviors of humans to conduct multi-model collaboration (Khan et al., 2024; Liang et al., 2023; Du et al., 2023; Yin et al., 2023; Sun et al., 2023). The goal is to combine the advantages of multiple models and introduce external feedback from other models to break through the performance bottleneck of a single model (Liang et al., 2023). The simplest method of multi-model collaboration is to perform majority voting based on the answers given by multiple LLMs. In addition, many methods begin to allow LLMs to interact with each other. Liang et al. (2023) and Du et al. (2023) enhanced the performance of LLM in specific tasks by allowing multiple LLMs to debate on the same problem and finally reach a consensus. Yin et al. (2023) proposed a cross-model exchange based on network topology to obtain feedback from other LLMs to improve their own output. Inspired by human behavior, Sun et al. (2023) proposed multiple collaboration modes, including discussion, review, and retrieval, to jointly work towards enhancing model inference performance. Different from these methods that interact at the overall answer level, our proposed TTE collaborates at the lowest token level and directly generates answers by reaching consensus at each token, avoiding possible tedious multi-round discussions.

## 5 CONCLUSION

In this paper, we propose TTE, an innovative token-level multi-model collaboration paradigm. By selecting the optimal next token from the next token distributions given by multiple LLMs and generating answers autoregressively, TTE can effectively integrate the capabilities of multiple models to break the performance bottleneck of a single model. The excellent performance on multiple tasks proves the effectiveness and reliability of our method. In addition, compared with previous multi-model collaboration methods, TTE has other advantages, such as no need to manually construct prompts for collaboration, no need for multiple rounds of discussion, and even shows certain emergence capabilities. We hope that TTE can provide a new perspective for multi-model collaboration. Discussion and limitation are provided in Appendix C.

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

## A  THEORETICAL ANALYSIS OF WHY TTE IS EFFECTIVE

The knowledge of a large language model (LLM) is primarily stored in its vast number of parameters, which encode a broad understanding of language and domain-specific knowledge. However, the external manifestation of this knowledge is the next token probability distribution generated at each step of autoregressive sampling. Formally, given a context $x_{<t}$, an LLM generates a probability distribution over the next possible tokens:

$$P_{\text{LLM}}(w_t \mid x_{<t}) = \text{softmax}(f_\theta(x_{<t})),$$

where $f_\theta$ represents the internal computation of the LLM parameterized by $\theta$. This probability distribution reflects the LLM's internal knowledge and informs the quality of the generated answers. Higher-quality knowledge results in a higher probability assigned to tokens that contribute to better answers. The final output from the model is sampled from this next token distribution, and the sequence of sampled tokens constitutes the generated text.

The answer space derived from this autoregressive process is defined by the cumulative sampling across multiple steps. Let the space of possible answers be denoted as $A$, with each potential answer having an associated probability based on the product of next-token probabilities. Hence, the probability of an answer $a \in A$ being generated by the LLM is:

$$P_{\text{LLM}}(a \mid x) = \prod_{t=1}^{T} P_{\text{LLM}}(w_t \mid x_{<t}),$$

where $T$ is the length of the generated answer. Importantly, the better the model's knowledge, the higher the probability assigned to higher-quality answers in this space. Theoretically, LLM can output any answer, but since some answers have extremely low probability of occurrence, we believe that LLM is not capable of making certain answers at this time. We believe that the answer space only contains answers with probability reaching a certain threshold.

Combining Knowledge from Multiple LLMs.

We propose to improve the overall answer quality by combining the knowledge of multiple LLMs, leveraging their individual next token distributions. Let $P^{(i)}(w_t \mid x_{<t})$ represent the next-token probability distribution generated by the $i$-th LLM. By combining these distributions, we create a new, enhanced distribution that incorporates the knowledge encoded in multiple models. Specifically, we aggregate the distributions as:

$$P_{\text{combined}}(w_t \mid x_{<t}) = \text{Aggregate}(P^{(1)}(w_t \mid x_{<t}), P^{(2)}(w_t \mid x_{<t}), \ldots, P^{(K)}(w_t \mid x_{<t})),$$

where $K$ is the number of models and the aggregation function is designed to effectively combine the distributions. The resulting answer space from this combination is strictly larger and of higher quality than any individual model's answer space, as it benefits from the union of knowledge across models. In theory, any answer contained in the single model answer space can be sampled from this answer space.

For example, consider two LLMs, $M_A$ and $M_B$, answering two different questions. In some cases, $M_A$ may provide the correct answer, while in other cases, $M_B$ might be more accurate. By merging their answer spaces, we can capture the correct answers from both models with higher probability, ensuring that:

$$P_{\text{combined}}(a^* \mid x) \geq \max(P^{(A)}(a^* \mid x), P^{(B)}(a^* \mid x)),$$

where $a^*$ is the optimal answer. In this way, the combined space encompasses the high-quality answers from both models and assigns them higher probabilities than the individual models would on their own. This probability is from a general perspective, because the probability that the merged answer space contains the correct answer is definitely greater than the answer space of a single LLM.

Sampling from the Combined Answer Space.

To efficiently sample high-quality answers from this enhanced space, we design several sampling strategies that prioritize high-probability tokens and consistency across models. These methods are based on two principles: (1) maximizing the aggregated probability (the aggregation method can be cooperation or competition) of sampled tokens and (2) ensuring consistency in token selections across models.

For instance, one approach is to sample the token with the highest aggregate probability at each step:

$$w_t = \arg\max_w P_{\text{combined}}(w \mid x_{<t}),$$

Alternatively, we can introduce multi-model consensus-based sampling, where a token is selected only if it is ranked highly across multiple LLMs:

$$w_t = \text{Consensus}(\{w_t^{(i)} \mid i = 1, \ldots, K\}),$$

where $w_t^{(i)}$ is the top token sampled from the $i$-th LLM's distribution. This ensures that only tokens which multiple models agree on are selected, further increasing the likelihood of generating high-quality answers.

In conclusion, by combining the next-token distributions of multiple LLMs and introducing sophisticated sampling strategies, we create a richer and more reliable answer space that enhances the probability of producing high-quality answers. The combined knowledge from multiple models overcomes the limitations of any single model and significantly improves the robustness and performance of generated responses.

## B  MORE EXAMPLES

Table 8: Examples where only one of the three single LLMs answers correctly. Incorrect answers are shown in red and correct answers are shown in blue. Since some answers are too long, we have made some adjustments to facilitate presentation, but the core content remains unchanged.

|  | GSM8K | CSQA |
|---|---|---|
| **Question** | *John runs 60 miles a week. He runs 3 days a week. He runs 3 hours the first day and half as much the other two days he runs. How fast does he run?* | *Aside from water and nourishment what does your dog need? (A) bone (B) charm (C) petted (D) lots of attention (E) walked* |
| **Qwen** | John runs 3 hours on the first day, and half as much on the other two days. This means he runs 3/2 = 1.5 hours on each of the other two days. So in total, he runs 3 + 1.5 + 1.5 = 6 hours per week. He runs 60 miles in 6 hours, so his speed is 60/6 = 10 miles per hour. The answer is: 10 | Analysis: Dogs, like all animals, require various needs to maintain their health and happiness. While water and nourishment are essential, dogs also need physical activity, mental stimulation, and affection. Among the given options, "walked" (Option E) is directly related to the physical exercise that dogs need. Answer: (E) walked |
| **Llama** | John runs 60 miles a week. He runs 3 days a week. So he runs 60 / 3 = 20 miles a day. He runs 3 hours the first day, and half as much the other two days. So he runs 3 hours the first day, and 1.5 hours each of the other two days. 3 + 1.5 + 1.5 = 6 hours. So he runs 20 miles in 6 hours. 20 / 6 = 3.33 miles per hour. The answer is 3.33. | Analysis: Dogs need a variety of things to stay happy and healthy, aside from water and nourishment. One of the most important things they need is exercise, which is closely related to option (E) walked. Correct answer: (E) walked |
| **GLM** | The total distance he runs in a week is 60 miles, and the total time he runs is: 3 hours (first day) + 3 hours (second day) + 3 hours (third day) = 9 hours Now, we can calculate his average speed for the week: 60 miles / 9 hours = 6.67 miles per hour So, John's average speed for the week is approximately 6.67 miles per hour. | Analysis: To determine the correct answer, we need to consider the basic needs of a dog, which include physical, emotional, and social requirements. Answer: (D) lots of attention |
| **TTE** | John runs 60 miles a week. He runs 3 days a week. On the first day, he runs 3 hours. On the other two days, he runs half as much as the first day, which is 3/2 = 1.5 hours each day. So, the total hours he runs in a week is: 3 hours (first day) + 1.5 hours (second day) + 1.5 hours (third day) = 6 hours To find out how fast he runs, we divide the total miles he runs by the total hours he runs: 60 miles / 6 hours = 10 miles per hour John runs at a speed of 10 miles per hour. | Analysis: Dogs are social animals that require attention, exercise, and affection to maintain their physical and mental health. While bones can be a treat, they are not a necessity for a dog's overall well-being. Charm is not a requirement for a dog's needs. Petting can be a form of affection, but it is not a fundamental need. Walking is essential for a dog's physical health and exercise, which is a crucial aspect of their care. Answer: (D) lots of attention |

Table 9: Examples where two of the three single LLMs answer correctly. Incorrect answers are shown in red and correct answers are shown in blue. Since some answers are too long, we have made some adjustments to facilitate presentation, but the core content remains unchanged.

|  | GSM8K | CSQA |
|---|---|---|
| **Question** | *Josh decides to try flipping a house. He buys a house for $80,000 and then puts in $50,000 in repairs. This increased the value of the house by 150%. How much profit did he make?* | *An underrated thing about computers is how they manage workflow, at one time it was a big deal when they could first do what? (A) share files (B) do arithmetic (C) turn on (D) cost money (E) multitask* |
| **Qwen** | Josh bought a house for $80,000 and spent $50,000 on repairs, so the total cost of the house is $80,000 + $50,000 = $130,000. The value of the house increased by 150% of its original value. The original value of the house was $80,000, so the increase in value is 150/100 * $80,000 = $120,000. This means that the new value of the house is $80,000 + $120,000 = $200,000. To find the profit, we subtract the total cost from the new value of the house: $200,000 - $130,000 = $70,000. The answer is $70000. | Analysis: The question is asking about an underrated aspect of computers that was once considered a significant achievement. Among the options, "multitasking" (option E) is often seen as an advanced feature that wasn't necessarily a big deal in the early days of computing. However, considering the context of the question, which seems to focus on early milestones, the most fitting answer would be "do arithmetic" (option B). In the early days of computers, their ability to perform basic arithmetic operations was indeed a significant achievement, as it demonstrated their computational capabilities beyond simple data processing or file management. Answer: (B) do arithmetic |
| **Llama** | Josh bought the house for 80,000. He put in 50,000 in repairs. So the total cost is 80,000 + 50,000 = 130,000. The house increased in value by 150%. So the new value is 130,000 * 2.5 = 325,000. The profit is the difference between the new value and the cost. 325,000 - 130,000 = 195,000. The answer is 195,000. | Analysis: The question is asking about an underrated thing about computers, which is their ability to manage workflow. The correct answer is the option that was a big deal when computers first started doing it. Multitasking was a significant advancement in computer technology, as it allowed computers to perform multiple tasks simultaneously, making them more efficient and productive. Correct answer: (E) multitask |
| **GLM** | To calculate the profit Josh made, we first need to determine the new value of the house after the repairs. The value of the house increased by 150%, which means the increase is 150% of the original value of the house. To find this, we calculate: 150% of $80,000 = 1.5 * $80,000 = $120,000 This $120,000 is the amount by which the value of the house increased. To find the new value of the house, we add this increase to the original value: New value of the house = Original value + Increase New value of the house = $80,000 + $120,000 New value of the house = $200,000 Now, to find the profit, we subtract the total cost (original purchase price + repair costs) from the new value of the house: Profit = New value of the house - (Original purchase price + Repair costs) Profit = $200,000 - ($80,000 + $50,000) Profit = $200,000 - $130,000 Profit = $70,000 Josh made a profit of $70,000. | Analysis: The question is asking about an "underrated thing about computers" that was a significant development when computers first had the capability to do it. Let's analyze each option: (A) share files: While sharing files is a crucial aspect of computer networking, it was not the first significant development when computers were first introduced. (B) do arithmetic: Computers have always been capable of doing arithmetic, and this was not a groundbreaking development when they were first introduced. (C) turn on: The ability to turn on a computer is a fundamental feature, but it is not something that would be considered an "underrated thing about computers" or a significant development when computers were first created. (D) cost money: The fact that computers cost money is a given, and it is not a feature that would be considered a significant development or an "underrated thing." (E) multitask: Multitasking refers to the ability of a computer to perform multiple tasks simultaneously. This was a significant development when computers first gained this capability, as it allowed for more efficient use of the system's resources and improved productivity. Based on the analysis, the correct answer is: (E) multitask |
| **TTE** | To calculate the profit Josh made, we first need to determine the new value of the house after the repairs. The value of the house increased by 150%, which means the new value is 150% of the original value plus the original value itself. The original value of the house is $80,000. The increase in value is 150% of $80,000, which is: 150% of $80,000 = 1.5 * $80,000 = $120,000 So, the new value of the house after the repairs is: $80,000 (original value) + $120,000 (increase) = $200,000 Josh spent $50,000 on repairs, so the total cost of the house after repairs is: $80,000 (original purchase price) + $50,000 (repairs) = $130,000 Now, to calculate the profit, we subtract the total cost from the new value of the house: $200,000 (new value) - $130,000 (total cost) = $70,000 Josh made a profit of $70,000. | Analysis: The question is asking about an underrated aspect of computers and a significant milestone in their development. The correct answer should be something that was a major achievement in the early days of computing. (A) Share files: While file sharing is an important feature, it's not a groundbreaking achievement that was a big deal in the early days of computers. (B) Do arithmetic: This is a fundamental capability of computers, but it's not something that was particularly groundbreaking or a big deal in the early days of computing. (C) Turn on: This is a basic function of any electronic device, not a significant achievement in the development of computers. (D) Cost money: This is not a feature or capability of computers, but rather a factor that affects their accessibility. (E) Multitask: This is a significant achievement in the development of computers, as it allows them to perform multiple tasks simultaneously, which was not possible in the early days of computing. Answer: (E) Multitask |

Table 10: Examples where all of the three single LLMs answer correctly. Incorrect answers are shown in red and correct answers are shown in blue. Since some answers are too long, we have made some adjustments to facilitate presentation, but the core content remains unchanged.

| | GSM8K | CSQA |
|---|---|---|
| **Question** | *Janet's ducks lay 16 eggs per day. She eats three for breakfast every morning and bakes muffins for her friends every day with four. She sells the remainder at the farmers' market daily for $2 per fresh duck egg. How much in dollars does she make every day at the farmers' market?* | *A revolving door is convenient for two direction travel, but it also serves as a security measure at a what? (A) bank (B) library (C) department store (D) mall (E) new york* |
| **Qwen** | Janet's ducks lay 16 eggs per day. She eats 3 eggs for breakfast, so she has 16 - 3 = 13 eggs left. She uses 4 eggs to bake muffins, so she has 13 - 4 = 9 eggs left to sell. She sells each egg for $2, so she makes 9 * $2 = $18 from selling eggs every day. The answer is: $18 | Analysis: Revolving doors are commonly used in public places to control access and maintain a certain level of security. They are particularly useful in areas where there is a need to control who enters and exits, such as banks, where sensitive transactions occur, and large buildings with high foot traffic. Answer: (A) bank |
| **Llama** | Janet lays 16 eggs per day. She eats 3 and bakes 4, so she has 16 - 3 - 4 = 9 eggs left. She sells these 9 eggs for 2 dollars each. 9 x 2 = 18. The answer is 18. | Analysis:A revolving door is a type of door that allows people to enter and exit through the same door, but in opposite directions. It is often used in high-security areas to control the flow of people and prevent unauthorized entry. In this context, the correct answer would be an option that is typically associated with high-security measures. Answer: (A) bank |
| **GLM** | Janet's ducks lay 16 eggs per day. She eats 3 eggs for breakfast, so she has 16 - 3 = 13 eggs left.
She bakes muffins for her friends using 4 eggs, so she has 13 - 4 = 9 eggs remaining.
She sells these 9 eggs at the farmers' market for $2 per egg, so she makes 9 * $2 = $18 every day at the farmers' market. | Analysis: A revolving door is designed to allow people to enter and exit in both directions smoothly, which can be particularly convenient in high-traffic areas. However, it also serves as a security measure by controlling the flow of people in and out of a building. This can be particularly important in places where security is a priority, such as financial institutions. Options Analysis:
(A) Bank - Banks are known for their security measures, and controlling access is crucial. A revolving door could serve as a security measure here.
(B) Library - While libraries may have security concerns, they are not typically as high as those in financial institutions.
(C) Department store - Security is important, but it is not as critical as in a bank.
(D) Mall - Similar to a department store, while security is important, it is not as critical as in a bank.
(E) New York - This is not a specific location where a revolving door would serve as a security measure.
Based on the analysis, the correct answer is: (A) bank |
| **TTE** | Janet's ducks lay 16 eggs per day. She eats 3 eggs for breakfast and uses 4 eggs for baking muffins. So, she uses a total of 3 + 4 = 7 eggs for personal use.
This leaves her with 16 - 7 = 9 eggs to sell at the farmers' market.
She sells each egg for $2, so she makes 9 * $2 = $18 every day at the farmers' market. | Analysis: A revolving door is a type of door that allows people to enter and exit in both directions, but it also serves as a security measure. This is because it can help to control the flow of people entering and exiting a building, making it more difficult for unauthorized individuals to enter. This is particularly useful in places where security is a concern, such as financial institutions. Answer: (A) bank |

Table 11: Examples where TTE answers incorrectly. Incorrect answers are shown in red and correct answers are shown in blue. Since some answers are too long, we have made some adjustments to facilitate presentation, but the core content remains unchanged.

| | GSM8K | CSQA |
|---|---|---|
| **Question** | *Elliott is trying to walk 10,000 steps a day. He finished half of his steps on his walks to and from school and did another 1,000 steps going for a short walk with his friend. He also went for a short jog around the block and realized that after he had finished his jog, he only had 2,000 steps left to take. How many steps did Elliott take during his jog?* | *What would vinyl be an odd thing to replace? (A) pants (B) record albums (C) record store (D) cheese (E) wallpaper* |
| **Qwen** | Elliott finished half of his steps on his walks to and from school, which means he took 10,000/2 = 5,000 steps on those walks. He also took 1,000 steps going for a short walk with his friend. So far, he has taken 5,000 + 1,000 = 6,000 steps. After his jog, he still had 2,000 steps left to take. This means that the number of steps he took during his jog is the total number of steps he needs to take (10,000) minus the steps he has already taken (6,000) and the steps he still needs to take (2,000). So, the number of steps he took during his jog is 10,000 - 6,000 - 2,000 = 2,000. The answer is: 2000. | Analysis: Vinyl is a type of material that was commonly used in the production of record albums and, less frequently, for certain types of wallpaper. It is not typically associated with clothing or food items like cheese. Therefore, the most unlikely option for replacing something with vinyl would be (D) cheese. Answer: (D) cheese |
| **Llama** | Elliott started with 10,000 steps. He did half of that, which is 5,000 steps. Then he did another 1,000 steps. So he did 5,000 + 1,000 = 6,000 steps. He had 2,000 steps left. So he must have done 6,000 + 2,000 = 8,000 steps. The answer is 8,000. | Analysis: The word "vinyl" is often associated with records, as vinyl records are a type of phonograph record made of polyvinyl chloride (PVC). Therefore, it would be an odd thing to replace something that is not related to records. Correct answer: (B) record albums |
| **GLM** | Elliott has a goal of 10,000 steps a day. He has already completed half of his steps on his walks to and from school, which is 10,000 / 2 = 5,000 steps.
He also did another 1,000 steps going for a short walk with his friend, so in total, he has completed 5,000 + 1,000 = 6,000 steps.
After his jog, he has 2,000 steps left to take, which means he has taken 10,000 - 2,000 = 8,000 steps so far.
To find out how many steps he took during his jog, we subtract the steps he took before the jog from the total steps he has taken so far: 8,000 - 6,000 = 2,000 steps.
Therefore, Elliott took 2,000 steps during his jog. | Analysis: The term "vinyl" is commonly associated with a few specific items. It is a type of plastic material that has been used in various applications over the years. Here's a brief analysis of each option: (A) pants - While "vinyl pants" are not common, they do exist, and vinyl can be used in the production of certain types of clothing. So, it's not an odd thing to replace pants with vinyl. (B) record albums - Vinyl is the material traditionally used for record albums, so replacing them with vinyl would not be odd. (C) record store - A record store is a place where vinyl records are sold, so replacing a record store with vinyl would be odd because vinyl is a material, not a place. (D) cheese - Cheese is a food product, and vinyl is a material. Replacing cheese with vinyl would be odd because they are completely different types of items. (E) wallpaper - Vinyl can be used to make wallpaper, so replacing wallpaper with vinyl is not an odd thing. Based on the analysis, the odd thing to replace would be: (D) cheese |
| **TTE** | Elliott has a goal of 10,000 steps a day. He has already completed half of his steps on his walks to and from school, which is 5,000 steps. He also did another 1,000 steps going for a short walk with his friend. So, he has completed 5,000 + 1,000 = 6,000 steps so far. He has 2,000 steps left to reach his goal of 10,000 steps. This means that the steps he took during his jog must be the difference between the steps he has completed and the steps he has left to complete his goal.
So, the steps he took during his jog are 10,000 - 6,000 = 4,000 steps. | Analysis: Vinyl is a type of material that is often used to make records, which are used to play music. It is also used in some other applications, such as in the production of certain types of clothing and accessories. However, it is not typically used to make pants, cheese, or wallpaper. Therefore, the odd thing to replace with vinyl would be something that is not typically made from vinyl. Answer: (A) pants |

We show more specific examples in Table 8, Table 9, Table 10 and Table 11, corresponding to examples where TTE correctly answered questions when only one, two, and three single models answered correctly, and examples where TTE answered incorrectly. Combined with Table 7, we can find that TTE can effectively aggregate the capabilities of multiple models. As long as one member can give the correct answer, TTE has the ability to give the correct answer, and can even give the correct answer when all member models give incorrect answers.

However, TTE can also make mistakes. As can be seen from Table 11, TTE can also give wrong answers when some member models can answer correctly. Moreover, it cannot always exhibit emergent capabilities when none of the member models answer correctly. This also shows that TTE still has room for improvement and its potential has not been fully explored. This is also the goal of our further research.

## C DISCUSSION AND LIMITATION

TTE is a method that leverages multi-model collaboration to improve model performance from the inference side. It can break through the performance bottleneck of a single model without training and without consuming a large amount of resources. Compared to previous multi-model collaboration methods, TTE does not require manually designing prompts to enable collaboration among multiple models, nor does it involve the hassle of multiple rounds of interaction to reach a consensus. A single round of autoregression is sufficient to produce a complete and deterministic answer.

Furthermore, we believe that TTE can be used for high-quality annotation, as it can integrate the knowledge boundaries of multiple models to provide high-quality pseudo-labels for unlabeled data. The annotation quality is likely to be superior to that of a single model and could help address the current shortage of high-quality data.

Notably, we must also pay attention to whether this multi-model collaboration approach might bypass some of the restrictions of single-model generated answers, leading to the production of unethical or harmful content. This is an area that requires further research in our future work.

