# OpenReview forum: "Token-by-Token Election: Improving Language Model Reasoning through Token-Level Multi-model Collaboration"
_ICLR.cc/2025/Conference — Submitted to ICLR 2025_

### Official Review · Reviewer_GGPR · 2024-10-27

**Soundness:** 2
**Presentation:** 2
**Contribution:** 2
**Rating:** 3
**Confidence:** 4

**Summary:**

This paper proposes a new multi-model collaboration strategy called Token-by-Token Election (TTE), aiming to enhance the reasoning abilities of language models through word-level collaboration. The TTE method achieves improvements in tasks such as mathematical reasoning, commonsense reasoning, and reliability assessment.

**Strengths:**

1. Introduction of a novel token-level multi-model collaboration strategy (TTE). TTE is the first to adopt a token-level strategy in multi-model collaboration, offering a new perspective in this field. It introduces three election modes—cooperative, competitive, and counting—all aimed at selecting the optimal token from multiple distributions, thereby strictly controlling the quality of each step in the autoregressive generation. It demonstrates excellent performance across multiple reasoning benchmarks.

2. No need for manual prompt construction in collaboration. Unlike other multi-model collaboration methods, TTE does not require manual prompt construction or multi-round discussions, simplifying the collaborative process.

3. In certain cases, TTE can provide the correct answer even when all individual models produce incorrect responses, showcasing an emergent ability.

**Weaknesses:**

1. Limitations of the multi-model collaboration approach. Although TTE can surpass the performance of individual models in certain scenarios, it still relies on most models providing the same correct answer or on a single model providing a high-confidence correct answer to persuade other models to reach a consensus.

2. Need for further exploration of emergent abilities. TTE does not consistently demonstrate emergent abilities across all scenarios, and its generalizability across different tasks remains unclear.

3. Lack of distinction from previous multi-model collaboration methods. The paper needs to more clearly elaborate on its technical contributions to differentiate TTE from existing multi-model collaboration methods.

**Questions:**

1. Could you provide more examples of different types of tasks that demonstrate TTE's emergent abilities?
2. How is knowledge represented in the definition of KN on LINE-147-148?
3. How does TTE ensure coherence in generation across the three collaborative methods?
4. When using token-level collaboration by different models' tokenizers, how does TTE ensure the generation of complete words or phrases?

**Details Of Ethics Concerns:**

None.

---

### Official Review · Reviewer_AzUF · 2024-11-01

**Soundness:** 2
**Presentation:** 3
**Contribution:** 2
**Rating:** 3
**Confidence:** 5

**Summary:**

This paper presents a token-level multi-model collaboration strategy. It selects the optimal next token from the candidate next token distributions by cooperation, competition and counting. Experiments on mathematical tasks, commonsense reasoning and symbolic reasoning are conducted to verify the effectiveness of the method.

**Strengths:**

1. Good motivation.
2. The method is simple and straightforward. It is easy for the readers to reproduce.

**Weaknesses:**

1. Authors overclaimed the contributions "we are the first to attempt multi-model collaboration at the token level, offering a new perspective in this field." There are some existing work, such as DeePen (arXiv:2404.12715) and GaC (arXiv:2406.12585). Both of them are probability-level (token-by-token) model ensembling and should not be considered as contemporaneous works (before July 1). More recently, Unite (arXiv:2410.03777) also works on this direction whereas it was posted to arxiv after the DDL of ICLR. Just for kindly reminder. Authors should discuss the differences between TTE and DeePen/GAC and compare with them.

2. Honstly, I do not think the TTE method is reasonable and can work well. A notable problem is how to align the different vocabulary and tokenizer. For example, the word "llama" is tokenized to "lla" and "ma" with Model1 while to "ll" and "ama" with Model2. Actually, both Model1 and Model2 are going to generate the word "llama" but their immediate next words are different owing to the tokenization method. In such cases, cooperation and counting would fail. Therefore, I think TTE cannot essentially address the core problem, i.e. aligning vocabulary and tokenizations, in probability-level model collaboration. At least, DeePen, GaC and Unite try to solve this problem.

3. Lack of experiments on larger language models, such as model >30B, since GaC indicates that model ensembling performs better with larger models.

**Questions:**

1. Instead of only selecting one static strategy during one inference time, could these three strategies can dynamically selecting accoding to the candidate next token probabilities?

---

### Official Review · Reviewer_Z94S · 2024-11-02

**Soundness:** 2
**Presentation:** 3
**Contribution:** 3
**Rating:** 5
**Confidence:** 3

**Summary:**

This paper studies how to combine the predictions of multiple large language models (LLMs) at the token level at the inference time. It proposes a token-by-token election (TTE) method, which combines the predictions from multiple LLMs. This paper includes three variants of TTE, each differs by how the predictions are combined. First, Cooperation works by creating an expanded vocabulary and summing the probability of top H words from each LLM. Second, Competition works by selecting the token with the highest probability among all LLMS. Third, Counting, similar to Cooperation, selects the token with the highest count instead of drawing from the summed probability distribution.

Experiments are conducted on various tasks including arithmetic reasoning (e.g., GSM8K), commonsense reasoning (e.g., CommonseQA), symbolic reasoning, and TurthfulQA. The base LLMs (to be combined) are Qwen2-7B, Llama3-8B, and GLM4-9B. In the main results, the TTE variants are compared against single LLM performance, CoT, and majority voting where the paper demonstrates that TTE outperforms these baselines on the majority of tasks.

**Strengths:**

Multi-model/system/agent and inference-time methods are of interest to the community given that training LLMs are becoming more expensive. The proposed method, TTE, shows superior performance.

**Weaknesses:**

1. *Limited baseline comparison*: The paper mentions two other multi-model collaboration methods (EoT and MAD in line 265). However, there is limited discussion on how TTE is different or better compared to these baselines. Section 3.5 simply reports experimental results without any discussion or in-depth analysis of the strengths and weaknesses of these various strategies.

2. The proposed method requires access to prediction distributions – this means that it won’t be applicable to models with API access (i.e. text only outputs). Thus, in practice, TTE requires, for example, K models to be hosted concurrently. In this case, Qwen + Llama + GLM, but it is not clear to me whether it would be more effective to run, for example, a 30B model which would require a similar inference infrastructure.

3. *The soundness of the method*: The competition variant in particular simply selects the token from the model that is most confident. I agree with the statement in lines 202-203 that “the higher the probability, the greater its confidence…”. However, these LLMs are “not” calibrated, and confidence may not correlate with correctness/truthfulness, e.g., models can be highly confident yet incorrect – and this is a well-known problem in deep learning. I wonder if the authors have insights or comments on this issue.

**Questions:**

The points I have in weaknesses can also be considered my questions. Can the authors make comments on the weaknesses?

---

### Official Review · Reviewer_dxMG · 2024-11-03

**Soundness:** 2
**Presentation:** 2
**Contribution:** 2
**Rating:** 3
**Confidence:** 4

**Summary:**

This paper proposes token-by-token election (TTE), token-level multimodel collaboration strategy. It selects the optimal token from the token distributions given by multiple LLMs and generates the answer based on the combination of those. The authors tried multiple token election approaches (cooperation, competition and counting)

**Strengths:**

The idea of this paper is simple, even if used before in multiple ensembles of seq2seq models, but the different ways for multiple token election approaches (cooperation, competition and counting) shows valuable insights to the reader.

**Weaknesses:**

Missing related work:

The paper fails to cite related work in the area (see e.g., https://arxiv.org/pdf/2403.03870)

The idea of ensembling the prediction of multiple sequence to sequence models (like LLms), has been done in many papers in prior work. Starting from these ones in 2010 decade:

1. Ilya Sutskever, Oriol Vinyals, and Quoc V. Le. 2014. Sequence to sequence learning with neural networks. In Proc. of Neurips.
2. Oriol Vinyals, Lukasz Kaiser, Terry Koo, Slav Petrov, Ilya Sutskever, and Geoffrey Hinton. 2015. Grammar as a foreign language. In Proc. of Neurips.
3. Kuncoro et al. Distilling an Ensemble of Greedy Dependency Parsers into One MST Parser. EMNLP.

**Questions:**

The abstract claims "due to their inherent limitations, the performance of a single model on many complex reasoning tasks has reached a bottleneck" - do we believe this to be true? from my point of view, models are getting better at many tasks and even multiple modalities. This does not invalidate the motivation of the paper, but is there something you could cite that shows this ? The paper you are citing in the intro is from 2022, I think the reasoning capabilities of LLMs have improved by a wide margin in the last 2 years.

is there a way to make this work with models that do not share the same vocabulary? any thoughts on this?

re: competition. Line 200. Would it be a good idea to weight that by the model scores on relevant benchmarks? Not just by model "confidence" based on scoring distribution? If a model is too confidence, you may end up with lower scores

The results do not show a clear winner among the different approaches in all tasks. What would be your recommendation for future work that want to reimplement what you present in the paper?

---

### Meta-Review · Area_Chair_8bVY · 2024-12-06

**Metareview:**

This paper proposes token-by-token election (TTE), a "token-level method" for generating text with multiple autoregressive sequence-to-sequence models. The work considers three variants of this approach—cooperation, counting, and competition—to ensemble model probability estimates. The experimental results on reasoning and QA benchmarks show that TTE outperforms the provided baselines when used to ensemble Qwen-2, Llama-3, and GLM-4.

Strengths:
- Ensembling generative models is becoming an important topic, as training new models is increasingly expensive, and the proposed approach is a reasonable and straightforward method for achieving this, even on models with different vocabularies.
- The approach demonstrates good performance in the provided experimental settings.

Weaknesses:
- One of the main weaknesses of this work is that the writing is unclear, which makes it difficult to understand the contributions and findings. For example, the discussion in Section 2.2 makes it unclear if the models are being ensembled at the token level (as the method's name suggests) or at the word level. If the ensembling is word-level, some concerns raised by the reviewers would be addressed—but details explaining *how* this word-level sampling is performed, and why the method is still considered "token-level" throughout the rest of the paper, are not provided.
- The experimental results provide limited insights into how the method works and do not discuss why the three proposed approaches perform best in different settings. Many of the concerns here are due to missing experiments, such as no baseline comparisons on TruthfulQA (whereas they are provided on the reasoning tasks) and limited analyses.

Other important weaknesses raised by the reviewers include:
- Missing discussion of related prior work, which also extends to limited comparisons to these works
- Limited consideration of the efficiency of this method, as it requires hosting the ensembled models simultaneously (or having access to the full model probabilities through an API, which is very uncommon).

**Additional Comments On Reviewer Discussion:**

The reviewers all raised points to discuss with the authors, but the authors did not participate in the response period.

---

### Decision · Program_Chairs · 2025-01-22

Reject